# The impact of assisted reproductive technologies versus natural conception on neonatal intensive care unit admission: A retrospective cohort analysis

Huajuan Chen[1], Hui Shao[2], Lei Xu[3], Xiujuan Wang[ID][2]*

1 Department of Delivery Room, Shaoxing Maternity and ChildHealth Care Hospital, Shaoxing, China,
2 Department of Infectology, Shaoxing Maternity and ChildHealth Care Hospital, Shaoxing, China,
3 Department of Gynecology and Obstetrics, Shaoxing Maternity and ChildHealth Care Hospital, Shaoxing, China

* juliewang472@163.com

## Abstract

### Background

The rapid advancement of assisted reproductive technology (ART), coupled with the increasing prevalence of advanced maternal age, has led to a global rise in ART-conceived neonates. Whether these reproductive methods affect neonatal health outcomes, particularly regarding the risk of neonatal intensive care unit (NICU) admission, has become a critical concern in perinatal medicine.

### Objective

This study aimed to compare the risk of NICU admission between neonates conceived naturally and those conceived through ART in a large cohort population, with important implications for optimizing perinatal care and improving neonatal outcomes.

### Methods

A retrospective study was conducted to analyze the baseline data of 3,867 singleton mothers and their neonates, who were either pregnant through ART or spontaneous conception, at a tertiary maternity hospital in Zhejiang Province from 2022 to 2024. Propensity score matching (PSM) was employed to match seven potential confounders that might affect the outcomes. Logistic regression analyses (univariate and multivariate, pre- and post-PSM) assessed the association between gestation mode and NICU admission risk, with additional multivariable-adjusted models for deeper investigation.Additionally, subgroup analyses were conducted pre- and post-PSM to explore how the mode of gestation impacts the risk of neonatal NICU admission in different population subsets. Finally, PSM was applied to five maternal factors (age,

**Data availability statement:** All relevant data are within the paper and its Supporting Information files.

**Funding:** The author(s) received no specific funding for this work.

**Competing interests:** The authors have declared that no competing interests exist.

BMI, gestational weeks, gravidity,parity, and pregnancycomplications). Differences in neonatal characteristics, such as gestational weeks, birth weight, delivery method, and delivery-related hemorrhage, were compared across different gestational modes pre- and post-PSM using box scatter plots.We also performed mediation analysis to assess the potential mediating effects of confounding factors, including the mode of delivery and gestational weeks.

## Results

Among 3,867 births, 265 neonates were admitted to the NICU. Restricted cubic spline logistic regression analyses demonstrated that ART-conceived neonates had a lower risk of NICU admission compared to naturally conceived neonates, both before and after PSM.This may be attributed to enhanced prenatal monitoring and selective embryo transfer in ART pregnancies, which could mitigate adverse perinatal outcomes. Subgroup analyses before PSM identified an interaction between cesarean section and ART, which was not observed after PSM.Overall, the results of the subgroup analyses suggest that neonates born through ART have a lower risk of NICU admission across various population subgroups. Box scatter plots showed that ART-conceived neonates had shorter gestational weeks, lower birth weights, higher cesarean section rates, and greater intrapartum hemorrhage (all $P < 0.05$), with no significant difference in sex distribution ($P > 0.05$).Additionally, the mediation analysis quantified the effect sizes mediated by delivery mode and gestational age.

## Conclusion

ART-conceived neonates have a reduced risk of NICU admission compared to naturally conceived neonates,potentially due to optimized prenatal care and embryo selection offsetting the risks associated with shorter gestation and lower birth weight.However, the elevated rates of cesarean delivery and intrapartum hemorrhage in ART pregnancies require ongoing clinical attention to improve maternal and neonatal outcomes. These findings suggest that while ART may confer neonatal benefits, it carries important maternal risks that warrant consideration in clinical decision-making.

## Introduction

The neonatal intensive care unit (NICU) is a specialized ward for critically ill newborns, and NICU admission is a key predictor of neonatal morbidity and mortality. With the rapid advancement of assisted reproductive technology (ART) and shifting socioeconomic factors, the number of ART-conceived newborns has risen globally. Concurrently, cesarean section rates remain high worldwide. It is estimated that over 8 million children have been born through ART, and this number continues to grow [1]. While natural conception remains the primary reproductive method, ART involves

interventions such as in vitro fertilization, embryo culture, and transfer, which may influence embryonic development and the maternal gestational environment, potentially impacting neonatal health [2].

ART pregnancies are often associated with closer prenatal monitoring, elective single embryo transfer (eSET) to reduce multifetal gestations, and higher socioeconomic status among ART users—all factors that may contribute to improved neonatal outcomes. Specifically, increased prenatal surveillance allows for earlier detection and management of complications, while eSET minimizes risks associated with preterm birth and low birth weight. Additionally, the demographic profile of ART users (e.g., higher education, better access to healthcare) may further mitigate adverse neonatal outcomes, potentially lowering NICU admission rates compared to naturally conceived pregnancies.

Whether different reproductive methods affect neonatal health, particularly the risk of NICU admission, has become a focal point in perinatal medicine. Numerous studies have explored the association between ART and NICU admission risk, but findings remain inconsistent. Some studies suggest ART is associated with a higher risk of NICU admission compared to natural conception [3–4], while others report no significant association [5–6]. These discrepancies may stem from variations in study design, sample size, and inadequate control for key confounding factors such as maternal age, parity, and pre-existing conditions.Propensity score matching (PSM) is particularly well-suited to address these limitations as it: (1) balances observed covariates between ART and naturally conceived groups, (2) mimics randomization in observational studies by creating comparable cohorts, and (3) allows simultaneous adjustment for multiple potential confounders that may differentially affect NICU admission risk.Notably, many studies fail to account for the potential benefits of ART-related practices (e.g., stringent embryo selection, optimized maternal care), which could explain the observed heterogeneity in results.

To address this, we conducted a large cross-sectional study to examine the impact of ART versus natural conception on adverse neonatal outcomes, with NICU admission as the primary outcome. Unlike previous studies, we used PSM to rigorously control for seven key maternal and obstetric confounders, thereby reducing selection bias and enabling more accurate estimation of the true association between conception method and NICU admission risk.The primary objective of this study was to determine whether ART conception is associated with differential risk of NICU admission compared to natural conception, while controlling for potential confounding factors through propensity score matching. Additionally, we aimed to explore potential mediating factors that might explain any observed associations.Our findings indicate that ART-conceived newborns have a reduced risk of NICU admission compared to naturally conceived neonates,likely mediated by the aforementioned advantages of ART protocols, but the high cesarean section rate associated with ART warrants ongoing attention.

## Methodology

### Date of access to data

Data accessed on 1 February 2024 and authors had access to information that could identify individual participants during or after data collection.

### The study population

The study population comprised 3,867 singleton mothers admitted to our hospital between January 2022 and December 2024, along with 3,867 newborns conceived either through assisted reproductive technology or natural pregnancy. Fig 1 illustrates the flowchart for population screening and study inclusion.This retrospective study was approved by the Ethics Committee of Shaoxing Maternal and Child Health Hospital in January 2023, the ethical approval number is [IRB-AF-022–01.5]. And it did not involve animal or human clinical trials. And data collection was based on complete medical records and data analysis was performed anonymously. The ethics committee waived the requirement for informed consent. All our research methods were in accordance with relevant guidelines and regulations.

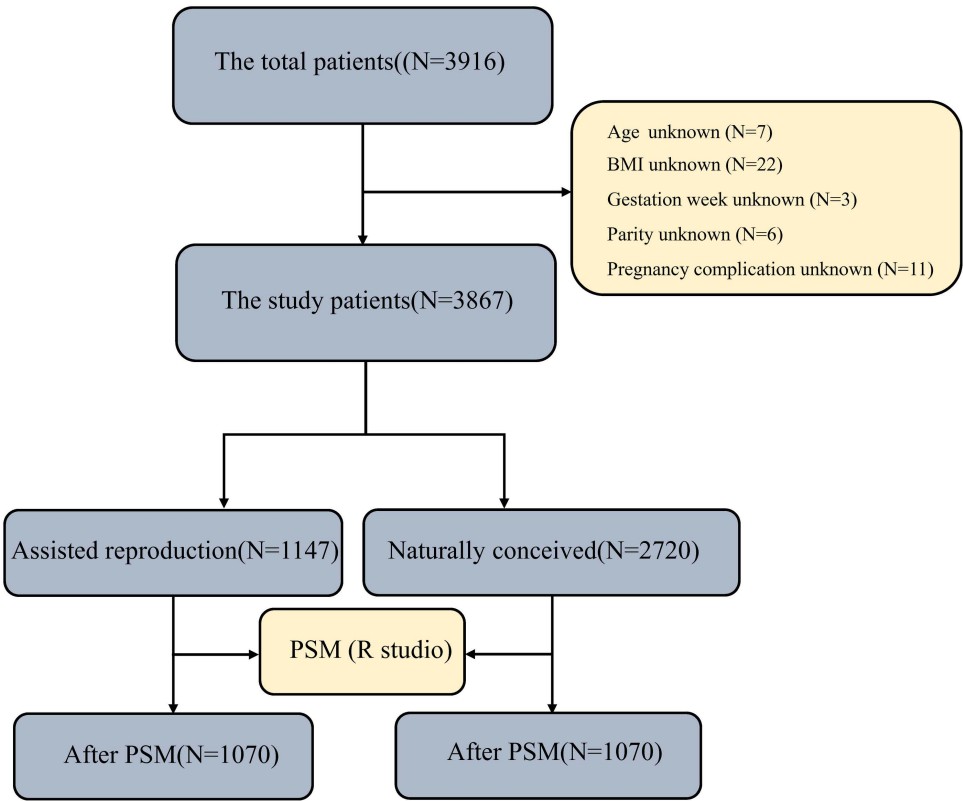

**Fig 1. Flowchart of participant screening and study inclusion process.**

## Data collection

We conducted a retrospective review of maternal and neonatal characteristics using data extracted from electronic medical records and nursing documentation systems. This study was a retrospective analysis based on existing medical data, with no direct intervention or additional data collection from the study subjects.The variables collected included maternal age, BMI, gestational age, gravidity, parity,pregnancy-related complications(specifically hypertensive disorders of pregnancy and gestational diabetes mellitus), mode of conception (natural or ART), gestational week at delivery, mode of delivery (vaginal or cesarean section), newborn sex, birth weight, and NICU admission status. These factors were carefully reviewed and validated by a multidisciplinary team comprising experienced obstetricians, midwives, and neonatologists to ensure data reliability and accuracy.In our study, the term ART encompasses the following specific procedures: in vitro fertilization (IVF), intracytoplasmic sperm injection (ICSI), frozen embryo transfer (FET), ovarian induction (OI), intrauterine insemination (IUI), and blastocyst transfer. However, it should be noted that while our dataset includes all these ART modalities, the medical records did not consistently specify the exact type of ART procedure for each case. As a result, our analysis treats ART as a collective variable rather than examining outcomes by specific ART subtypes. We acknowledge this as a methodological limitation that prevents us from conducting more granular analyses of differential outcomes across various ART techniques. This limitation primarily stems from the retrospective nature of our data collection from clinical records.The admission criteria for the NICU in our study followed a standardized institutional protocol that included perinatal asphyxia, significant birth trauma (e.g., clavicular fracture or intracranial hemorrhage), respiratory disorders (including pneumonia, pulmonary complications, or need for mechanical ventilation), confirmed or suspected sepsis, very

low birth weight (<1500 grams) or gestational age < 32 weeks, major congenital malformations or other congenital conditions requiring immediate intervention (e.g., congenital heart defects), and conditions secondary to intrauterine infection.

## Statistical analysis

We conducted descriptive analyses to characterize all participants in the study. Continuous variables were expressed as mean ± standard deviation (SD) and analyzed using independent t-tests, while categorical variables were expressed as percentages and analyzed using chi-square tests.To minimize confounding bias, we performed propensity score matching (PSM) in two stages: First, we conducted primary PSM (1:1 ratio) using seven key covariates (maternal age, BMI, gestational age, parity, gravidity, pregnancy comorbidities, and delivery mode) for our main outcome analysis. Second, we performed secondary PSM focusing on five relatively stable maternal characteristics (age, BMI, gravidity, parity, and pregnancy comorbidities) to generate comparative visualizations of other clinical variables. Additionally, we constructed a directed acyclic graph (DAG) to clarify our causal assumptions in the mediation analysis, which is presented in Supplementary S2 Fig.The DAG specifically accounts for gestational age and delivery mode as variables that may serve dual roles in our analysis – both as potential mediators in the causal pathway between ART conception and NICU admission, and as important confounders that could independently influence both the exposure and outcome.For mediation analysis,we employed a Bayesian mediation analysis approach.This method widely used in observational studies to enhance statistical power and reduce bias [7–8]. Univariate and multivariate logistic regression analyses were performed before and after PSM to examine the relationship between reproductive mode and the risk of NICU admission. Restricted cubic spline models were utilized to further explore this association. Subgroup analyses were conducted based on maternal age, BMI, gestational age, parity, gravidity, pregnancy comorbidities, gestational week, and delivery mode. Furthermore, PSM was applied to five maternal factors (age, BMI, gravidity, parity, and pregnancy comorbidities), and box scatter plots were generated to compare gestational weeks, birth weights, sex, delivery methods, and intrapartum hemorrhage between groups before and after PSM. A p-value < 0.05 was considered statistically significant.Data were statistically analyzed using R software (version 4.4.1).

## Results

### Baseline Characteristics of the Study Population Before and After PSM

The study included 3,867 participants, and their baseline characteristics were compared between ART and natural conception groups using independent t-tests for continuous variables and chi-square tests for categorical variables. Table 1 summarizes the baseline characteristics of maternal and neonatal confounders according to the mode of conception. Before propensity score matching (PSM), mothers in the ART group were significantly older (p < 0.05), had higher BMI (p < 0.05), shorter gestational weeks (p < 0.05), were more likely to be primigravida, and had higher cesarean section rates (p < 0.05) compared to those with natural pregnancies. After PSM (1:1), no significant differences were observed in the seven confounders between the ART and natural conception groups (p > 0.05). Fig 2A and 2B display the distribution of data density and standardized mean differences (SMDs) before and after matching.Detailed SMDs values are provided in Supplementary S3 Table.

### Association between mode of conception and risk of NICU admission

Table 2 displays the results of univariate and multivariate logistic regression analyses, with neonatal NICU admission as the primary outcome. Univariate logistic regression was employed to assess the association between maternal age, BMI, gestational week, gravidity, parity, pregnancy comorbidities, delivery mode, and the risk of NICU admission. Before PSM, gestational week was inversely associated with NICU admission risk [0.67 (0.63–0.71)], while higher parity [0.59 (0.38–0.92)] and prior birth experience [0.55 (0.37–0.83)] were linked to reduced NICU admission risk.Vaginal delivery was

**Table 1. Baseline Characteristics of the Study Population Before and After Propensity Score Matching.**

| Variable | Before PSM | | | P | After PSM | | | P |
|---|---|---|---|---|---|---|---|---|
| | Total (n = 3867) | Naturally con-ceived (n = 2720) | Assisted repro-duction (n = 1147) | | Total (n = 2140) | Naturally con-ceived (n = 1070) | Assisted repro-duction (n = 1070) | |
| Age, Mean ± SD | 30.33 ± 4.33 | 29.70 ± 4.18 | 31.84 ± 4.31 | **<.001** | 31.39 ± 4.17 | 31.33 ± 4.29 | 31.45 ± 4.05 | 0.518 |
| BMI, Mean ± SD | 22.22 ± 3.71 | 22.03 ± 3.50 | 22.67 ± 4.12 | **<.001** | 22.51 ± 3.92 | 22.49 ± 3.73 | 22.53 ± 4.11 | 0.810 |
| Gestation week, Mean ± SD | 38.80 ± 1.75 | 38.88 ± 1.75 | 38.62 ± 1.75 | **<.001** | 38.64 ± 1.89 | 38.62 ± 2.05 | 38.66 ± 1.72 | 0.584 |
| Gravidity, n (%) | | | | 0.150 | | | | 0.601 |
| ≤2 | 2776 (71.79) | 1971 (72.46) | 805 (70.18) | | 1515 (70.79) | 763 (71.31) | 752 (70.28) | |
| >2 | 1091 (28.21) | 749 (27.54) | 342 (29.82) | | 625 (29.21) | 307 (28.69) | 318 (29.72) | |
| Parity, n (%) | | | | **<.001** | | | | 0.367 |
| Primipara | 2547 (65.87) | 1715 (63.05) | 832 (72.54) | | 1513 (70.7) | 747 (69.81) | 766 (71.59) | |
| Multipara | 1320 (34.13) | 1005 (36.95) | 315 (27.46) | | 627 (29.3) | 323 (30.19) | 304 (28.41) | |
| Pregnancy complica-tion, n (%) | | | | 0.401 | | | | 0.787 |
| No | 2429 (62.81) | 1697 (62.39) | 732 (63.82) | | 1376 (64.3) | 691 (64.58) | 685 (64.02) | |
| Yes | 1438 (37.19) | 1023 (37.61) | 415 (36.18) | | 764 (35.7) | 379 (35.42) | 385 (35.98) | |
| Delivery, n (%) | | | | **<.001** | | | | 0.892 |
| Cesarean Section | 1942 (50.22) | 1164 (42.79) | 778 (67.83) | | 1399 (65.37) | 698 (65.23) | 701 (65.51) | |
| Vaginal Delivery | 1925 (49.78) | 1556 (57.21) | 369 (32.17) | | 741 (34.63) | 372 (34.77) | 369 (34.49) | |

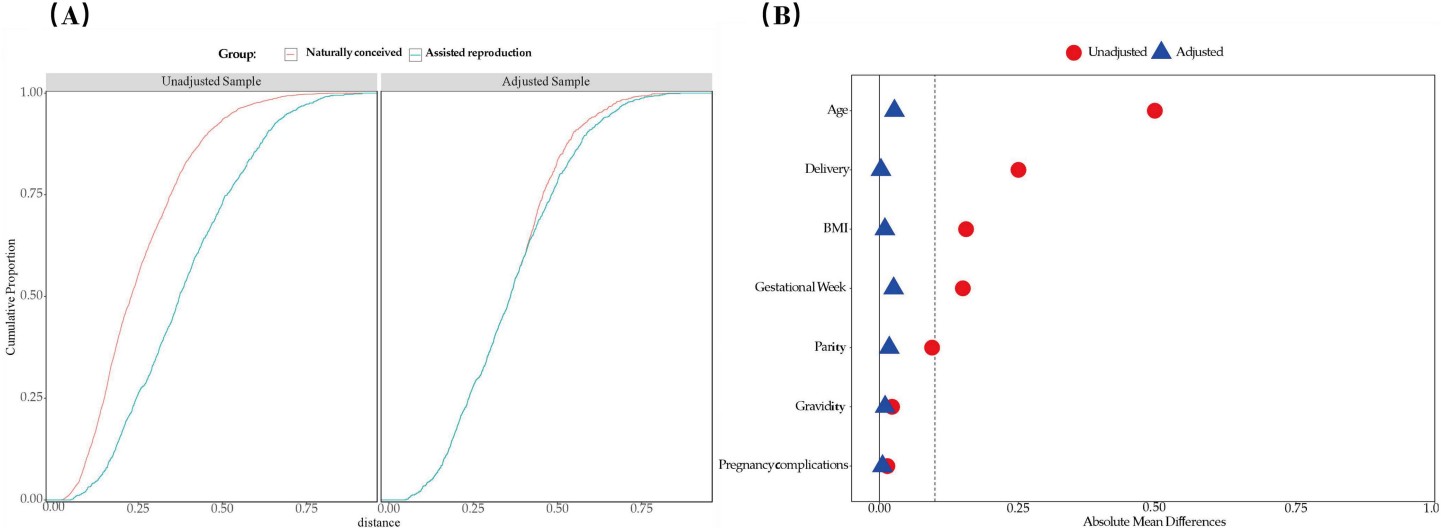

**Fig 2. 2A illustrates the data distribution before and after PSM, while 2B presents the standardized mean difference (SMDs) values before and after matching.** The results provided an intuitive visualization of balance improvement across all variables, clearly demonstrating that all post-PSM SMDs were below 0.1.

associated with an increased risk [5.77 (4.01–8.31)], and ART pregnancies had a lower NICU admission risk compared to natural pregnancies [0.64 (0.45–0.92)]. After PSM, gestational week remained inversely associated with NICU admission risk [0.70 (0.65–0.75)], vaginal delivery continued to increase risk [5.95 (3.76–9.43)], and ART pregnancies still exhibited a lower NICU admission risk [0.65 (0.44–0.97)].

**Table 2. Univariate and multivariate logistic regression analysis of different reproductive methods and the risk of newborns admitted to NICU.**

| Variable | Before PSM | | | | After PSM | | | |
|---|---|---|---|---|---|---|---|---|
| | Univariate logistic | | Multivariable logistic | | Univariate logistic | | Multivariable logistic | |
| | OR (95%CI) | P | OR (95%CI) | P | OR (95%CI) | P | OR (95%CI) | P |
| Age, Mean±SD | 0.95 (0.92~0.98) | **<.001** | 1.01 (0.98~1.05) | 0.507 | 0.97 (0.93~1.01) | 0.144 | | |
| BMI, Mean±SD | 1.00 (0.97~1.04) | 0.901 | | | 1.02 (0.99~1.07) | 0.225 | | |
| Gestation week, Mean±SD | 0.73 (0.69~0.76) | **<.001** | 0.67 (0.63~0.71) | **<.001** | 0.72 (0.68~0.77) | <.001 | 0.70 (0.65~0.75) | **<.001** |
| Gravidity, n (%) | | | | | | | | |
| ≤2 | 1.00 (Reference) | | 1.00 (Reference) | | 1.00 (Reference) | | 1.00 (Reference) | |
| >2 | 0.51 (0.37~0.71) | **<.001** | 0.59 (0.38~0.92) | **0.021** | 0.57 (0.37~0.89) | 0.014 | 0.55 (0.30~1.02) | 0.057 |
| Parity, n (%) | | | | | | | | |
| Primipara | 1.00 (Reference) | | 1.00 (Reference) | | 1.00 (Reference) | | 1.00 (Reference) | |
| Multipara | 0.46 (0.34~0.63) | **<.001** | 0.55 (0.37~0.83) | **0.004** | 0.51 (0.32~0.81) | 0.004 | 0.76 (0.43~1.37) | 0.369 |
| Pregnancy complication, n (%) | | | | | | | | |
| No | 1.00 (Reference) | | 1.00 (Reference) | | 1.00 (Reference) | | 1.00 (Reference) | |
| Yes | 1.51 (1.17~1.94) | **0.001** | 0.91 (0.69~1.22) | 0.533 | 2.07 (1.45~2.97) | <.001 | 1.05 (0.69~1.61) | 0.811 |
| Delivery, n (%) | | | | | | | | |
| Cesarean Section | 1.00 (Reference) | | 1.00 (Reference) | | 1.00 (Reference) | | 1.00 (Reference) | |
| Vaginal Delivery | 4.41 (3.24~6.00) | **<.001** | 5.77 (4.01~8.31) | **<.001** | 5.15 (3.48~7.64) | <.001 | 5.95 (3.76~9.43) | **<.001** |
| Method of conception,n (%) | | | | | | | | |
| Naturallyconceived | 1.00 (Reference) | | 1.00 (Reference) | | 1.00 (Reference) | | 1.00 (Reference) | |
| Assisted reproduction | 0.53 (0.39~0.73) | **<.001** | 0.64 (0.45~0.92) | **0.015** | 0.60 (0.42~0.87) | 0.007 | 0.65 (0.44~0.97) | **0.035** |

### Restricted multiple-model logistic regression analysis

Three logistic regression models were constructed to analyze the relationship between mode of conception and neonatal NICU admission risk, as shown in Table 3. Model 1 is an unadjusted model that does not account for covariates. Model 2 adjusted for maternal age, BMI, and pregnancy comorbidities. Model 3 further adjusted for gestational week, gravidity, parityand delivery mode. In all models, ART pregnancies were associated with a significantly lower risk of NICU admission compared to natural pregnancies, with ORs and 95% CIs of 0.53 (0.39–0.73), 0.58 (0.42–0.80), and 0.63 (0.44–0.91), respectively (p<0.05). This association remained consistent and was strengthened after matching, with ORs and 95% CIs of 0.60 (0.42–0.87), 0.60 (0.41–0.87), and 0.64 (0.43–0.96), p<0.05.

### Subgroup analysis conducted before and after PSM

To investigate the relationship between mode of conception and neonatal NICU admission risk, subgroup analyses were conducted, stratified by maternal age, BMI, gestational week, gravidity, parity, pregnancy comorbidities, and delivery mode (Figs 3A and 3B). Before PSM, cesarean section was found to significantly influence the association between mode of conception and NICU admission risk (interaction p<0.05).This initial association likely reflected residual confounding, as ART pregnancies often have higher cesarean rates due to clinician caution with precious pregnancies or maternal factors like advanced age. After PSM, none of the seven confounders showed a significant interaction with this association (interaction p>0.05). The attenuation of interaction effects post-PSM suggests that: (1) the original cesarean section interaction was partially confounded by imbalanced baseline characteristics between groups, and (2) propensity matching successfully controlled for these variables, revealing a more direct association.Both pre- and post-PSM subgroup analyses revealed that ART-conceived neonates had a lower risk of NICU admission compared to naturally conceived neonates (OR < 1).The consistent protective effect across subgroups (range: OR 1.37–2.78) suggests ART may confer benefits

**Table 3. Multi-Model Logistic Regression Analysis of of different reproductive methods and the risk of newborns admitted to NICU.**

| Model | Method of conception | Before PSM | | After PSM | |
|---|---|---|---|---|---|
| | | OR (95%CI) | P | OR (95%CI) | P |
| Model 1 | Naturallyconceived | 1.00 (Reference) | | 1.00 (Reference) | |
| | Assisted reproduction | 0.53 (0.39~0.73) | **<.001** | 0.60 (0.42~0.87) | **0.007** |
| Model 2 | Naturallyconceived | 1.00 (Reference) | | 1.00 (Reference) | |
| | Assisted reproduction | 0.58 (0.42~0.80) | **<.001** | 0.60 (0.41~0.87) | **0.006** |
| Model 3 | Naturallyconceived | 1.00 (Reference) | | 1.00 (Reference) | |
| | Assisted reproduction | 0.63 (0.44~0.91) | **0.013** | 0.64 (0.43~0.96) | **0.030** |

Model1: Crude.

Model2: Adjust: Age, BMI, Pregnancy complication.

Model3: Adjust: Age, BMI,Gravidity, Parity, Pregnancy complication, Gestation week, Delivery.

**Fig 3. Subgroup analyses were conducted to examine the association between pregnancy mode and the risk of neonatal NICU admission.** 3A displays the subgroup analysis before PSM, while 3B presents the subgroup analysis after PSM.

through mechanisms like enhanced prenatal monitoring or selective embryo transfer. However, the elevated OR for cesarean deliveries (2.78 post-PSM) indicates this subgroup warrants special clinical attention despite overall risk reduction.

## PSM was performed only for maternal age, BMI, gravidity, parity, and pregnancy-related comorbidities

Fig 4A and 4D compare gestational weeks between ART and natural pregnancy groups before and after PSM. Natural pregnancies had significantly longer gestational weeks compared to ART pregnancies (P < 0.05). Fig 4B and 4E compare neonatal birth weights, with the ART group showing slightly higher birth weights than the natural pregnancy group (P < 0.05). Fig 4C and 4F compare intrapartum hemorrhage, revealing that the ART group experienced greater bleeding during labor and delivery (P < 0.05). Fig 5A and 5B compare sex distribution between the two groups, showing no significant difference (P > 0.05). Fig 5C and 5D compare delivery modes, with the ART group exhibiting a higher cesarean

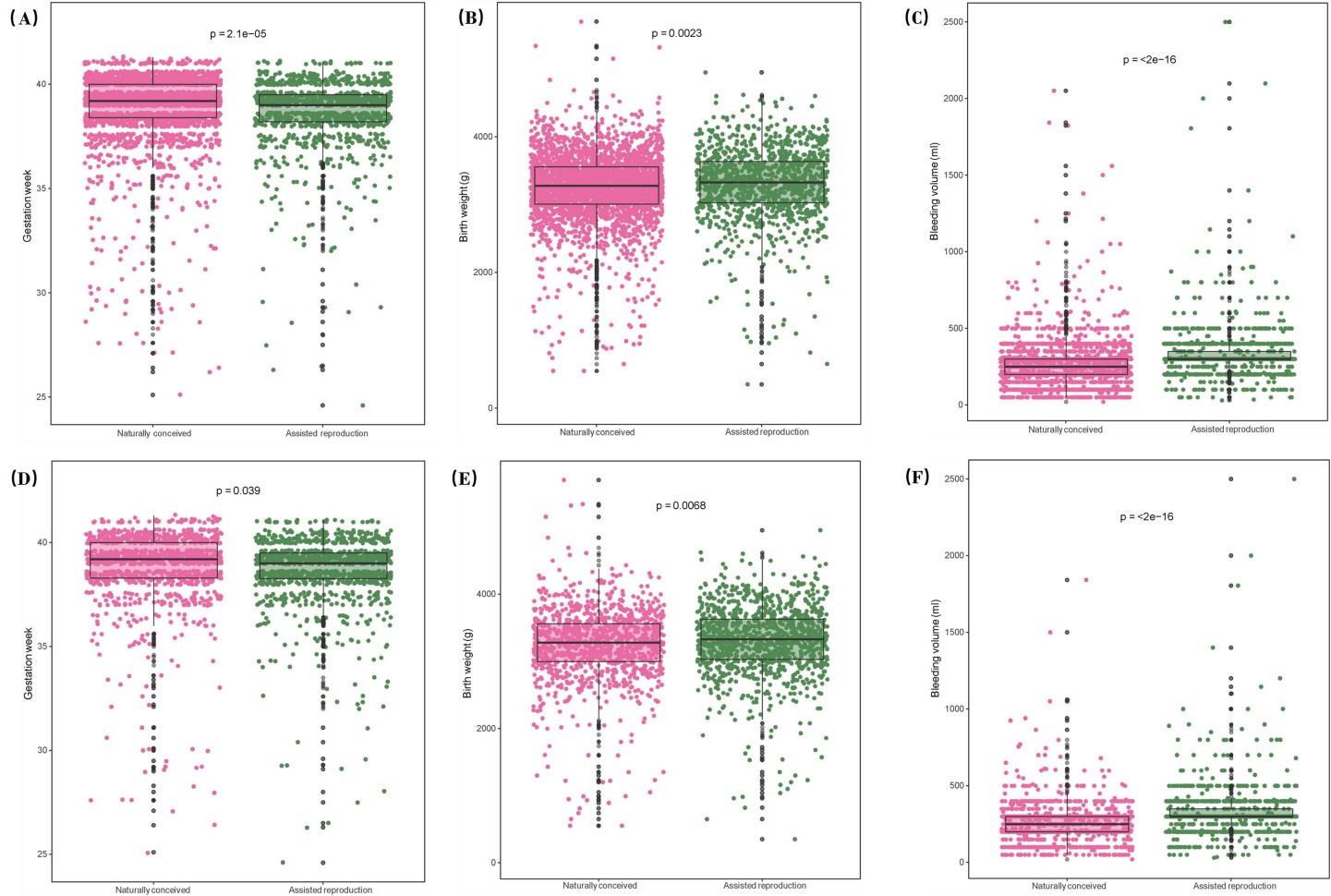

**Fig 4. Boxandwhisker plots depicting gestational weeks (4A, D), birth weight (4B, E), and bleeding during labor (4C, F)are shown for both ART pregnancies and natural pregnancies, before and after PSM.** Individual data points are shown as scatter dots, while the boxplot displays the inter-quartile range (IQR) with the lower boundary at Q1 (25th percentile), the central line at Q2 (median), and the upper boundary at Q3 (75th percentile).

section rate than the natural pregnancy group (P<0.05).Balance diagnostics showing the standardized mean differences (SMDs) after this round of propensity score matching are available in the Supplementary **S3** Table.

## Mediation effect analysis

In Fig 6, we present the mediation analysis examining the role of the mode of delivery and gestational weeks. The mode of delivery mediated 18.98% of the association between the method of conception and the risk of neonates being admitted to the NICU (Fig 6A), with an indirect effect of 0.006 (95% CI=0.004–0.01, P<0.05).This positive mediation suggests that the lower NICU admission risk in ART-conceived neonates is partially explained (≈19%) by higher cesarean delivery rates in this population, which may reflect more controlled delivery conditions and/or closer perinatal monitoring in ART pregnancies. In the mediation analysis for gestational weeks, the proportion mediated was −15.07% (Fig 6B), with an indirect effect of −0.003 (95% CI=−0.005 to −0.002, P<0.05).The negativemediation indicates gestational age acts as

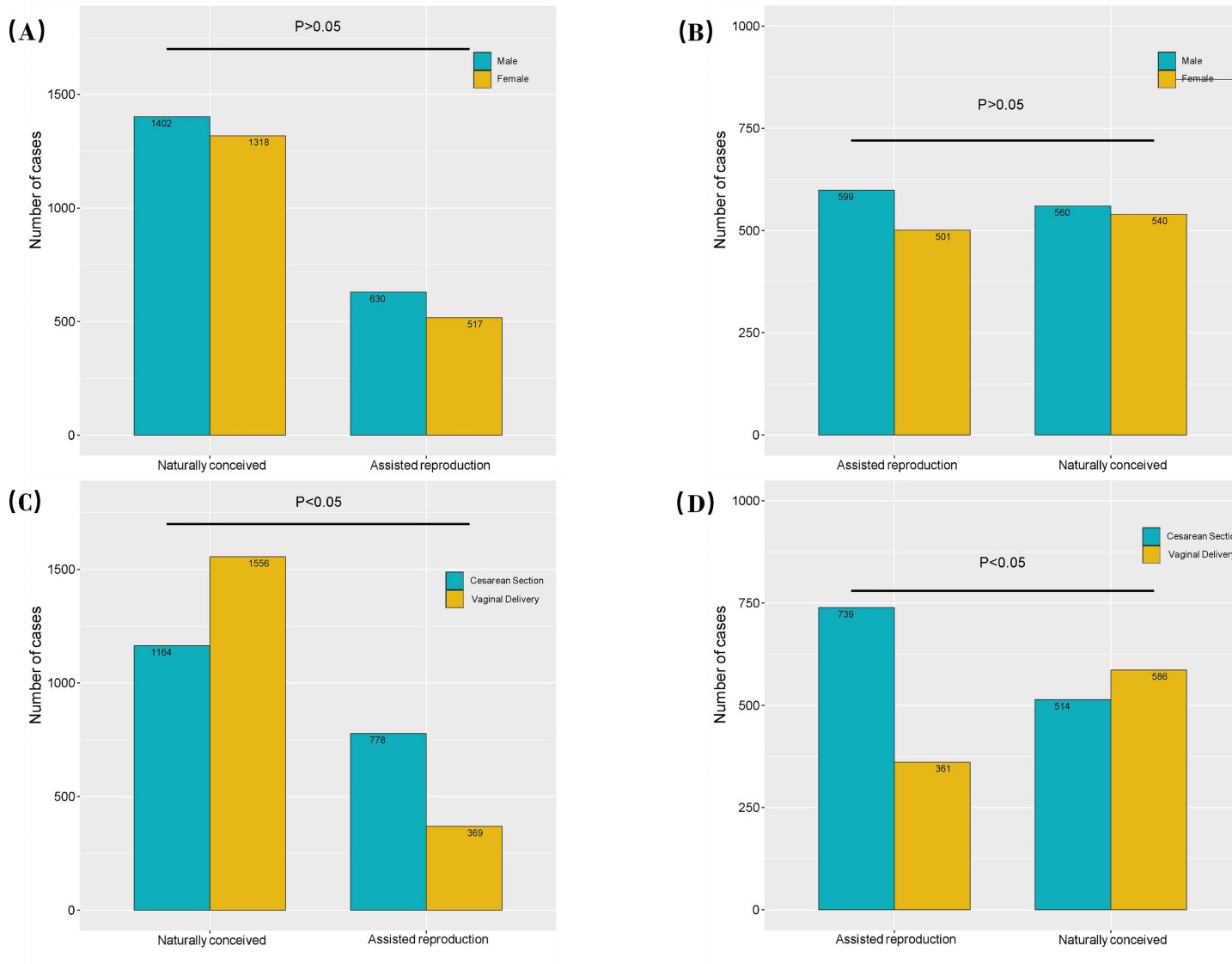

**Fig 5. A comparison of neonatal sex(5A, B) and mode of delivery(5C, D) between ART pregnancies and natural pregnancies is presented for both before and after PSM.** The height of the rectangle represents the number of individuals.

a suppressor variable: while ART is associated with slightly shorter gestation (mediated effect), this operates contrary to its overall protective effect, possibly because ART protocols select for healthier embryos that can better tolerate early delivery.

## Discussion

In this study, we compared the risk of NICU admission between neonates born from natural conception and those born through assisted reproductive technology (ART). Our results showed that ART-conceived neonates had a lower risk of NICU admission compared to those born from spontaneous conception. This finding contrasts with most previous studies, which have generally concluded that ART pregnancies may increase the risk of NICU admission [9–10]. The boxplot

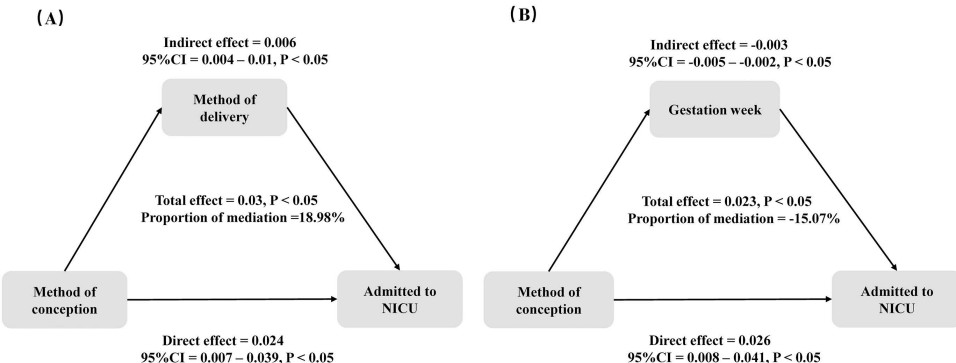

**Fig 6. Examining the relationship between the mode of delivery (6A) and gestational age (6B) as mediators of reproductive methods and their association with the risk of NICU admission.**

analyses showed that ART-conceived neonates had a lower risk of NICU admission despite having shorter gestational ages and lower birth weights – factors typically associated with higher NICU admission risk. This seemingly paradoxical finding can be explained by several compensatory mechanisms in ART pregnancies: (1) enhanced embryo selection through morphological and genetic screening may result in healthier embryos better able to tolerate shorter gestation; (2) intensive prenatal monitoring allows earlier detection and management of complications; and (3) the predominance of elective single embryo transfer eliminates risks associated with multiple gestations.

The mediation analysis revealed an unusual finding: the proportion mediated by gestational weeks was −15.07%, indicating a suppressor effect. This suggests that gestational weeks, typically a protective factor for NICU admission, acted in the opposite direction in our study. Clinically, this implies that the relationship between ART and reduced NICU admission risk is not solely explained by gestational age but may involve other unmeasured compensatory mechanisms, such as those described above. The negative mediation could also reflect residual confounding or complex interactions, such as the superior health status of ART-conceived neonates despite shorter gestation. This finding underscores the need for further research to disentangle the interplay between ART, gestational age, and neonatal outcomes.Although most studies have shown that ART pregnancies are associated with adverse perinatal outcomes, our study found a lower risk of NICU admission in ART-conceived neonates. This finding may be explained by several factors.First, ART pregnancies often involve closer prenatal monitoring and management. Women undergoing ART typically begin regular prenatal visits earlier in pregnancy, with more frequent ultrasounds and laboratory tests [11]. This comprehensive management facilitates early detection and treatment of potential complications, thereby reducing the risk of NICU admission.Second, as assisted reproductive technology has advanced and clinical guidelines have evolved, elective single embryo transfer (eSET) has become more common [12]. eSET effectively reduces the incidence of multiple pregnancies, a known risk factor for NICU admission [13].In this study, the ART pregnancy group consisted of women with singleton pregnancies, which further lowered the risk of NICU admission. Additionally, women undergoing ART tend to have higher socioeconomic status and greater health awareness [14], which may lead to better preconception care and health management during pregnancy. These women are more likely to avoid smoking and alcohol consumption, maintain a healthy weight, and exercise regularly—all factors that can positively influence neonatal health.Studies by Ginström, Wei, and Ishihara have shown that frozen embryo transfer can reduce the risk of preterm birth and low birth weight by improving embryo quality and endometrial receptivity, thereby further decreasing the likelihood of NICU admission [15–17]. Similarly, Wang's research indicated that optimizing ART techniques, such as personalized ovarian stimulation protocols, can improve neonatal outcomes [18]. Luke's study also highlighted that ART pregnancies typically involve closer prenatal monitoring, facilitating early detection and management of complications and consequently reducing NICU admissions [19]. These findings align with our results,

supporting the conclusion that ART pregnancies are associated with a reduced risk of NICU admission.However, we must acknowledge that assisted reproductive technologies are linked to higher rates of cesarean deliveries. Previous systematic reviews and meta-analyses have demonstrated that neonates delivered via cesarean section face an increased risk of birth asphyxia [20–21], which may elevate the likelihood of NICU admission. Given that NICU admission serves as a marker of adverse neonatal outcomes, our analyses revealed two key findings: First, within each delivery mode (vaginal or cesarean), ART-conceived neonates maintained lower NICU admission risks. Second, the magnitude of ART's protective effect was slightly attenuated but remained significant after accounting for delivery mode. This suggests that while cesarean delivery may partially mediate the ART-NICU admission relationship, other compensatory mechanisms (e.g., enhanced embryo selection, intensive prenatal monitoring) likely contribute substantially to the observed protective effect.

This study has several strengths and limitations. Notably, we employed propensity score matching (PSM) to control for confounding factors, enhancing the robustness of our findings. Additionally, we conducted stratified subgroup analyses to examine the relationship between mode of conception and NICU admission risk in different populations, highlighting the need for targeted prevention strategies. However, our study has limitations. First, despite controlling for major confounders, unmeasured variables(such as specific ART protocols, underlying causes of subfertility, subtle maternal health conditions not captured in our database, or unmeasured lifestyle factors) may have influenced the results. Second, the study only assessed short-term NICU admissions and did not evaluate the long-term health outcomes of ART neonates. Third, the single-center design may introduce selection bias and limit generalizability to other populations or healthcare settings. Future studies with larger cohorts and longer follow-up periods are needed to validate our findings and explore the differential effects of ART techniques, such as IVF and ICSI, on NICU admission risk and long-term neonatal health.

## Conclusions

The results of this study demonstrated that neonates conceived through assisted reproductive technology (ART) have a lower risk of NICU admission compared to those born from spontaneous conceptions. Nevertheless, the elevated rates of cesarean delivery and intrapartum hemorrhage in ART pregnancies require ongoing clinical attention to improve maternal and neonatal outcomes. These findings suggest that while ART may confer neonatal benefits, it carries important maternal risks that warrant consideration in clinical decision-making.Future multi-center studies with larger samples should validate these findings, including stratified analyses of ART protocols and long-term neonatal outcomes.

## Supporting information

**S1 File. Dataset.The complete dataset of this study.**
(CSV)

**S2 Fig. Figure DAG. Directed acyclic graph.**
(TIF)

**S3 Table. Table SMDs.Detailed SMDs values.**
(DOCX)

## Acknowledgments

We thank all participants and researchers involved in the study.

## Author contributions

**Conceptualization:** Huajuan Chen, Lei Xu, Xiujuan Wang.

**Data curation:** Huajuan Chen, Hui Shao, Lei Xu.

**Formal analysis:** Huajuan Chen, Hui Shao, Xiujuan Wang.

**Methodology:** Huajuan Chen, Hui Shao, Lei Xu, Xiujuan Wang.

**Writing – original draft:** Huajuan Chen, Hui Shao, Lei Xu.

**Writing – review & editing:** Xiujuan Wang.

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
