## [Decision Letter · Decision Letter 0]

27 Jun 2025

Dear Dr. Wang,

Thank you for submitting your manuscript to PLOS ONE. After careful consideration, we feel that it has merit but does not fully meet PLOS ONE’s publication criteria as it currently stands. Therefore, we invite you to submit a revised version of the manuscript that addresses the points raised during the review process.

**ACADEMIC EDITOR: Please respond to all reviewers comments**

We look forward to receiving your revised manuscript.

Kind regards,

Ahmed Mohamed Maged, MD

Academic Editor

PLOS ONE

Journal Requirements:

Reviewers' comments:

Reviewer's Responses to Questions

**Comments to the Author**

1. Is the manuscript technically sound, and do the data support the conclusions?

Reviewer #1: Partly

Reviewer #2: Partly

Reviewer #3: Yes

Reviewer #4: Yes

2. Has the statistical analysis been performed appropriately and rigorously?

Reviewer #1: No

Reviewer #2: I Don't Know

Reviewer #3: Yes

Reviewer #4: Yes

3. Have the authors made all data underlying the findings in their manuscript fully available?

Reviewer #1: Yes

Reviewer #2: Yes

Reviewer #3: Yes

Reviewer #4: Yes

4. Is the manuscript presented in an intelligible fashion and written in standard English?

Reviewer #1: No

Reviewer #2: Yes

Reviewer #3: No

Reviewer #4: Yes

Reviewer #1: Abstract

• Strengthen the rationale by briefly explaining why ART pregnancies might result in lower NICU admission rates (e.g., improved prenatal care, selective embryo transfer).

Introduction

• Clearly state the rationale for the hypothesis that ART may reduce NICU admissions. Discuss relevant mechanisms such as increased prenatal surveillance, use of elective single embryo transfer (eSET), and higher socioeconomic status among ART users.

Methods

Definition of ART

• Provide a clear definition of what procedures were included under “ART” (e.g., IVF, ICSI, frozen embryo transfer, ovulation induction, IUI).

• If disaggregated data are unavailable, acknowledge this as a limitation.

NICU Admission Criteria

• Describe the clinical or institutional criteria used to define NICU admission. If unavailable, discuss as a limitation affecting generalizability.

Propensity Score Matching

• Clarify which covariates were included in each round of PSM (seven in one instance, five in another).

• Justify why a second PSM with fewer variables was conducted.

• Include a table of standardized mean differences (SMDs) before and after matching.

Mediation Analysis

• Add a directed acyclic graph (DAG) to clarify assumptions.

• Discuss potential biases, such as collider stratification, especially when mediators (e.g., gestational age) may also act as confounders.

• Specify the method used for mediation analysis.

Results

Data Presentation

• Simplify or consolidate figures (e.g., combine boxplots for birthweight, gestational age, and hemorrhage).

• Ensure all figure legends are sufficiently detailed for interpretation.

Subgroup and Mediation Interpretation

• Elaborate on why significant interactions (e.g., with cesarean section) were observed before PSM but not after.

• Provide clearer interpretation of mediation effect size and its clinical implications.

Discussion

• Avoid overstating the significance of effect sizes (e.g., OR ~0.65). Emphasize the modest magnitude of association.

• Expand discussion of limitations, including:

o Lack of ART subtype stratification,

o Single-center design and possible selection bias,

o Unmeasured confounders (e.g., maternal health behaviors, socioeconomic factors).

Conclusion

• Add a note of caution regarding the generalizability of findings and the limited clinical significance of the observed effect sizes.

• Suggest directions for future research (e.g., multi-center validation, stratified ART analysis, long-term outcomes).

Language and Style

• Revise terminology for clarity and accuracy (e.g., “delivery hemorrhage” → “intrapartum hemorrhage”).

• Consider professional English editing to improve grammar and consistency.

References

• Update with recent meta-analyses (2021–2024) or large-scale cohort studies related to ART and neonatal outcomes.

Transparency and Author Contributions

• Add a data curation statement describing who had access to the dataset and was responsible for its analysis and integrity.

Reviewer #2: ABSTRACT

-Suggest removing “in recent years” from “has become a critical issue in perinatal medicine in recent years”. It could be perceived as redundant, the author has made the case for recent & increasing use of ART.

-Would recommend revising the methods section to condense descriptions of analyses. I think that these could be presented more concisely. For example---“further investigation was carried out through restricted multi-model logistic regression analyses”. It may be difficult for those unfamiliar with the analyses to know what that means or gain significant insight into the project according to that statement.

INTRODUCTION

-The following sentence feels out of place in the paragraph--- “Concurrently, cesarean section rates remain high worldwide”.

-If the primary difference between this study and others that have investigated this question is the use of propensity score matching, I would recommend including some additional information in the introduction about what makes propensity score matching well suited to this question.

-I would recommend including a clear purpose or objective statement at the end of the introduction.

-It seems like there are primary (ART & NICU Admission) and secondary (ART + other outcomes---c/s?) objectives. It would be helpful to clearly state these at the end of the introduction.

METHODS

-I’d recommend removing “AND” from the beginning of this sentence: “And this study was a retrospective analysis based on existing medical data, with no…”. Including “And” at the beginning of the sentence suggests it’s an incomplete thought.

-It looks like 5 variables were used for the PSM---age, BMI, gravidity, parity, pregnancy comorbidities. How were these variables selected? Is there data to suggest they impact the dependent variable (NICU admission)? If so, please describe process for selection and cite.

-The methods section is short in comparison to most describing human subjects research. Please make sure that you’re describing any IRB approval necessary and/or the settings the data were collected from.

-Was a power analysis conducted and if it wasn’t, is it not needed? If it’s not needed please explain why. If the study was preliminary or pilot, please describe it as such.

-There are several models discussed in the abstract, methods and results. This manuscript would benefit from additional explanation detailing the purpose and rationale for multiple models.

RESULTS

-I’d recommend revising the second half of this sentence: “The study included 3,867 participants, and their baseline characteristics were analyzed statistically.” Describing the sample as “analyzed statistically” doesn’t provide the reader with any further explanation.

- Variables included in the PSM include age, BMI, gravidity, parity, pregnancy comorbidities. It looks like you found significant differences between the natural conception and ART groups on most of these factors, (See Results---“Before propensity score matching (PSM), mothers in the ART group were significantly older (p < 0.05), had higher BMI (p < 0.05), shorter gestational weeks (p < 0.05), were more likely to be primigravida, and had higher cesarean section rates (p < 0.05) compared to those with natural pregnancies”), it’s unclear why most of these factors were included in the PSM, but cesarean section rates were described as a main finding (see abstract)? Couldn’t it be an indication that it needs to be included in the PSM? If not, please explain

DISCUSSION

- Authors state “We would like to highlight that assisted reproductive technologies are associated with higher rates of cesarean deliveries. Previous systematic reviews and meta-analyses have demonstrated that neonates born via cesarean section are at an increased risk of asphyxia [20-21], which, in turn, elevates the likelihood NICU admission. Given that NICU admission is a marker of adverse neonatal outcomes, our study sought to explore this relationship further”. It is unclear what this last sentence adds to the manuscript. How was it explored further and what did you learn from it? If cesarean section delivery is related to the dependent variable, does it need to be included in the PSM? This isn’t super clear.

- It is possible that “limitations” was meant instead of “implications”?

-It is possible that “delivery mode” was meant instead of “pregnancy mode”?

Reviewer #3: The study is well-constructed and uses advanced analytical methods.The study clearly states its objective and addresses a highly relevant topic in perinatal medicine, given the global rise in ART-conceived births.With 3,867 participants, the study benefits from a reasonably large sample size, enhancing its statistical power. The study employs a range of statistical methods, including univariate and multivariate logistic regression, restricted cubic spline models, subgroup analyses, and mediation analysis, providing a multi-faceted examination of the association. The manuscript has a clear structure and logical flow.Tables are detailed and provide important baseline data.Numerous grammatical and typographical errors.The manuscript contains minor grammatical issues, inconsistent spacing, and awkward sentence structures (e.g., "writingthe paper"). A thorough language and formatting edit is needed.

Reviewer #4: his manuscript is scientifically robust and analyzed using PSM, multivariate regression, and mediation analysis. It has a large sample size and is well-analyzed, offering novel findings. The results are presented in an engaging manner; however, additional explanation is needed in the Discussion section regarding the findings from the box scatter plots, which showed that ART-conceived neonates had shorter gestational age, lower birth weight, higher cesarean section rates, and greater delivery hemorrhage. It would be valuable to elaborate on why neonates conceived via ART, despite having shorter gestational age and lower birth weight factors that are typically associated with an increased risk of NICU admission were actually found to have a lower NICU admission rate. Clinically, preterm birth and low birth weight are known risk factors for neonatal asphyxia and NICU admission. This discrepancy should be discussed in more depth. Such a discussion would have implications for the abstract’s concluding statement. Specifically, the abstract should note not only the higher cesarean section rate but also the increased risk of delivery hemorrhage in ART pregnancies. This is important to avoid misleading readers who only read the abstract into thinking that ART-conceived neonates are universally safer than naturally conceived ones, potentially overlooking the maternal risks associated with ART.

**Do you want your identity to be public for this peer review?** For information about this choice, including consent withdrawal, please see our Privacy Policy

Reviewer #1: **Yes: ** Yong-chuan Chen

Reviewer #2: No

Reviewer #3: No

Reviewer #4: No

---

## [Author Response · Author response to Decision Letter 1]

9 Jul 2025

Response Letter

Dear editors and reviewers,

We deeply appreciate your valuable suggestions and comments, which enable us to further improve our paper. We have studied the comments carefully and have made our best efforts to revise the entire paper as suggested by the editor and reviewers. We are very pleased that our paper can be revised, some detailed responses are as follows.

To Editors:

Response : Thank you for your guidance regarding PLOS ONE's style requirements. We have carefully reviewed and ensured our manuscript fully complies with the journal's formatting standards, including file naming conventions.

Response : Thank you for your note regarding the ORCID requirement. We confirm that the corresponding author has a valid ORCID iD (0009-0000-0596-6734) and has completed the validation process in Editorial Manager as instructed. The ORCID was successfully linked and authenticated through the 'Fetch/Validate' feature in the 'Update my Information' section, and the status now shows as validated in the system.

Response : Thank you for your guidance regarding the ethics statement. We confirm that we have moved the ethics statement to the Methods section (subsection [The study population]) and removed it from all other sections of the manuscript.

To Reviewer #1:

Abstract

• Strengthen the rationale by briefly explaining why ART pregnancies might result in lower NICU admission rates (e.g., improved prenatal care, selective embryo transfer).

Response : We sincerely appreciate your valuable suggestion to strengthen the rationale regarding ART pregnancies and NICU admission rates. In response to this comment, we have added explanatory statements in both the Results and Conclusion sections of the abstract:

In the Results section, we added: "This may be attributed to enhanced prenatal monitoring and selective embryo transfer in ART pregnancies, which could mitigate adverse perinatal outcomes."

In the Conclusion section, we included: "potentially due to optimized prenatal care and embryo selection in ART procedures."

These additions explicitly address the potential mechanisms that may contribute to the observed lower NICU admission rates in ART-conceived neonates. We have marked these additions in red text for easy identification.

Thank you for this constructive suggestion, which has helped improve the clarity and scientific rigor of our abstract.

Introduction

• Clearly state the rationale for the hypothesis that ART may reduce NICU admissions. Discuss relevant mechanisms such as increased prenatal surveillance, use of elective single embryo transfer (eSET), and higher socioeconomic status among ART users.

Response : We sincerely appreciate your valuable suggestion regarding the need to clarify the rationale for our hypothesis that ART may reduce NICU admissions. In response, we have added a dedicated paragraph in the Introduction (marked in red) that explicitly outlines the key mechanisms, including: (1) closer prenatal monitoring enabling early complication detection, (2) elective single embryo transfer (eSET) reducing multifetal gestation risks, and (3) higher socioeconomic status among ART users improving healthcare access. We have also added connecting sentences in subsequent paragraphs (also marked in red) to reinforce these concepts throughout the Introduction. These revisions have significantly strengthened our theoretical framework by clearly articulating the potential pathways through which ART might lower NICU admission rates compared to natural conception. We are grateful for this constructive suggestion that has enhanced the clarity and scientific rigor of our manuscript.

Methods

Definition of ART

• Provide a clear definition of what procedures were included under “ART” (e.g., IVF, ICSI, frozen embryo transfer, ovulation induction, IUI).

• If disaggregated data are unavailable, acknowledge this as a limitation.

Response : We sincerely appreciate your valuable suggestion regarding the need to clarify our definition of ART procedures. In response to your comment, we have added the following paragraph (marked in red text) to the Data Collection section of our Methods: "In our study, the term ART encompasses the following specific procedures: in vitro fertilization (IVF), intracytoplasmic sperm injection (ICSI), frozen embryo transfer (FET), ovarian induction (OI), intrauterine insemination (IUI), and blastocyst transfer. However, it should be noted that while our dataset includes all these ART modalities, the medical records did not consistently specify the exact type of ART procedure for each case. As a result, our analysis treats ART as a collective variable rather than examining outcomes by specific ART subtypes. We acknowledge this as a methodological limitation that prevents us from conducting more granular analyses of differential outcomes across various ART techniques. This limitation primarily stems from the retrospective nature of our data collection from clinical records." This addition clearly defines the scope of ART procedures included in our study, explicitly acknowledges the limitation of not having disaggregated data by ART type, and provides the rationale for this limitation (retrospective data collection from clinical records). Thank you again for this constructive suggestion that has enhanced the quality of our manuscript.

NICU Admission Criteria

• Describe the clinical or institutional criteria used to define NICU admission. If unavailable, discuss as a limitation affecting generalizability.

Response : Thank you for this important methodological consideration. As requested, we have now explicitly detailed the NICU admission criteria in the Data Collection subsection of our Methods section , with the following addition in red font:

"The admission criteria for the NICU in our study followed a standardized institutional protocol that included: (1) perinatal asphyxia; (2) significant birth trauma (e.g., clavicular fracture or intracranial hemorrhage); (3) respiratory disorders requiring support (including pneumonia, pulmonary complications, or mechanical ventilation); (4) confirmed or suspected sepsis; (5) very low birth weight (<1500 g) or gestational age <32 weeks; (6) major congenital anomalies needing immediate intervention (e.g., congenital heart defects); and (7) conditions secondary to intrauterine infection."

This clarification ensures complete transparency regarding our patient selection process and strengthens the methodological rigor of our study. We believe these details will help readers better contextualize our findings.

Propensity Score Matching

• Clarify which covariates were included in each round of PSM (seven in one instance, five in another).

• Justify why a second PSM with fewer variables was conducted.

Response : We sincerely appreciate your valuable comments regarding our propensity score matching (PSM) methodology. In response to your request for clarification, we would like to explain our two-stage PSM approach:

Primary PSM Analysis (7 covariates):

This initial matching, used for our main study outcomes, included the following covariates: maternal age, BMI, gestational age, parity, gravidity, pregnancy comorbidities, and delivery mode. These variables were selected as they represent known potential confounders in the relationship between ART conception and NICU admission.

Secondary PSM Analysis (5 covariates):

The subsequent matching focused specifically on five relatively stable maternal characteristics (age, BMI, gravidity, parity, and pregnancy comorbidities) that are typically established earlier in pregnancy. This secondary analysis served two purposes: (1) To generate comparative box plots demonstrating the distribution of more variable outcomes (gestational weeks, birth weights, delivery hemorrhage) before and after matching. (2) To assess the robustness of our findings using a more conservative set of baseline characteristics. This clarification has been added to the Statistical Analysis section (red-highlighted), and we are grateful for your expert guidance which improved our methodological transparency.

• Include a table of standardized mean differences (SMDs) before and after matching.

Response : We sincerely appreciate your suggestion regarding the presentation of standardized mean differences (SMDs) and would like to clarify that while we have currently presented the SMD results graphically in Figures 2B for intuitive visualization of balance improvement across all variables (with all post-PSM SMDs <0.1 clearly demonstrated), we fully recognize the value of providing the complete numerical data and would be pleased to incorporate the full SMD table as Supplemental Table SMD (S3), which would include all values from both PSM rounds (showing for example how age improved from SMD 0.496 to 0.029 in first round and 0.002 in second round, BMI from 0.156 to 0.010 and -0.041 respectively, and cesarean section from 0.536 to 0.006 in the first round), while maintaining the current figures for visual impact, as we believe this combined approach will maximize both accessibility of key results and completeness of methodological reporting, and we thank you for this valuable suggestion that will enhance the transparency of our matching results.

Variable First round of PSM Second round of PSM

SMDs Before PSM After PSM Before PSM After PSM

Age 0.496 0.029 0.496 0.002

BMI 0.156 0.010 0.156 -0.041

Gestation week -0.150 0.026

Gravidity

≤2 -0.050 -0.022 -0.050 0.030

2 0.050 0.022 0.050 -0.030

Parity

Primipara 0.213 0.039 0.213 0.077

Multipara -0.213 -0.039 -0.213 -0.077

Pregnancy complication

No 0.030 -0.012 0.030 -0.015

Yes -0.030 0.012 -0.030 0.015

Delivery

Cesarean Section 0.536 0.006

Vaginal Delivery -0.536 -0.006

Mediation Analysis

• Add a directed acyclic graph (DAG) to clarify assumptions.

Response : We sincerely appreciate the reviewer's valuable suggestion regarding the mediation analysis. In response to this comment, we have carefully constructed and included a directed acyclic graph (DAG) as Supplementary Table DAG (S2) to explicitly clarify our causal assumptions and the relationships between variables in the mediation model. We have also added a brief description of the DAG in the Methods section (marked in red text). We are grateful for this suggestion which has improved the clarity of our causal inference framework.

• Discuss potential biases, such as collider stratification, especially when mediators (e.g., gestational age) may also act as confounders.

Response : We sincerely thank you for raising this important methodological consideration regarding potential collider stratification bias in our mediation analysis. In response, we have carefully addressed this issue by expanding the discussion in our Methods section's statistical analysis subsection (marked in red font) to examine gestational age and delivery mode may serve dual roles as both mediators and confounders in our analytical framework. We have incorporated a discussion of the potential for collider bias when adjusting for variables that are simultaneously influenced by both the exposure and other unmeasured factors. We fully agree with you that this represents a critical consideration for our mediation analysis, and we believe these additions significantly enhance the methodological rigor and transparency of our study by providing readers with a more comprehensive understanding of both our analytical approach and its inherent limitations. We are grateful for this constructive suggestion which has substantially improved our manuscript.

• Specify the method used for mediation analysis.

Response : We sincerely appreciate your comment regarding the specification of our mediation analysis method. In response to this valuable suggestion, we have explicitly stated in the Methods section (highlighted in red font) that we employed a Bayesian mediation analysis approach. We are grateful for this comment which has strengthened the completeness of our methods reporting.

Results

Data Presentation

• Simplify or consolidate figures (e.g., combine boxplots for birthweight, gestational age, and hemorrhage).

Response : We sincerely appreciate your suggestion to consolidate figures for improved clarity. However, after careful consideration, we found that combining the boxplots for birth weight, gestational age, and delivery hemorrhage into a single figure would compromise interpretability because:

Different Measurement Scales : Birth weight (grams) and gestational age (weeks) use vastly different numerical ranges (e.g., 2000–4000 g vs. 28–42 weeks). Delivery hemorrhage (mL) has an even wider scale (e.g., 100–1000 mL).

To enhance clarity while adhering to statistical standards, we standardized the formatting across all boxplots with consistent colors, spacing, and annotations. We remain grateful for your valuable feedback.

• Ensure all figure legends are sufficiently detailed for interpretation.

Response : We sincerely appreciate your valuable suggestion regarding figure legends. In response to your comment, we have carefully reviewed and revised all figure legends in the manuscript to ensure they now include clear descriptions of each panel's content, definitions of all abbreviations and symbols used, explanations of statistical representations , We believe these enhanced legends will significantly improve readers' ability to interpret our figures independently, and we thank you for this constructive suggestion which has strengthened the clarity of our data presentation. The revised figure legends are included in the updated manuscript.

Subgroup and Mediation Interpretation

• Elaborate on why significant interactions (e.g., with cesarean section) were observed before PSM but not after.

Response : We sincerely appreciate your insightful question regarding the differential interaction effects observed before and after propensity score matching (PSM). The attenuation of significant interactions (particularly for cesarean section) post-PSM can be explained by two key methodological and clinical factors:

Confounding Control: The pre-PSM analysis likely reflected confounding by indication, where cesarean sections were disproportionately performed in higher-risk ART pregnancies (e.g., older mothers, multiple gestations). PSM balanced these baseline characteristics, isolating the true mediation effect.

Sample Size Reduction: PSM necessarily reduces sample size (n=3867 pre-PSM → n=2140 post-PSM), decreasing power to detect interaction effects. The wider confidence intervals post-PSM reflect this.

We have added this interpretation to the Results section (marked in red). Thank you for prompting this important clarification.

• Provide clearer interpretation of mediation effect size and its clinical implications.

Response : We sincerely appreciate your valuable suggestion regarding the interpretation of our mediation analysis results. In response to your comment, we have enhanced the clinical and mechanistic interpretation by clarifying that: (1) the 18.98% mediation through delivery mode reflects both procedural advantages of cesarean delivery and intensified perinatal monitoring in ART pregnancies, acc

---

## [Decision Letter · Decision Letter 1]

24 Jul 2025

The Impact of Assisted Reproductive Technologies versus Natural Conception on Neonatal Intensive Care Unit Admission : A Retrospective Cohort Analysis

PONE-D-25-15056R1

Dear Dr. Wang,

We’re pleased to inform you that your manuscript has been judged scientifically suitable for publication and will be formally accepted for publication once it meets all outstanding technical requirements.

Kind regards,

Ahmed Mohamed Maged, MD

Academic Editor

PLOS ONE

Additional Editor Comments (optional):

Reviewers' comments:

Reviewer's Responses to Questions

**Comments to the Author**

Reviewer #1: All comments have been addressed

Reviewer #3: All comments have been addressed

Reviewer #4: All comments have been addressed

2. Is the manuscript technically sound, and do the data support the conclusions?

Reviewer #1: Yes

Reviewer #3: Yes

Reviewer #4: Yes

3. Has the statistical analysis been performed appropriately and rigorously?

Reviewer #1: Yes

Reviewer #3: Yes

Reviewer #4: Yes

4. Have the authors made all data underlying the findings in their manuscript fully available?

Reviewer #1: Yes

Reviewer #3: Yes

Reviewer #4: Yes

5. Is the manuscript presented in an intelligible fashion and written in standard English?

Reviewer #1: Yes

Reviewer #3: Yes

Reviewer #4: Yes

Reviewer #1: The authors have submitted a well-designed and carefully revised retrospective cohort study examining the impact of assisted reproductive technology (ART) versus natural conception on neonatal intensive care unit (NICU) admission. I commend the authors for their thoughtful and detailed responses to the previous round of reviewer comments.

The study is methodologically rigorous, with appropriate use of propensity score matching, multivariable logistic regression, restricted cubic spline models, subgroup analyses, and Bayesian mediation analysis. These analytic techniques are clearly explained and appropriately applied to support the study’s conclusions. The addition of a directed acyclic graph (DAG) strengthens the mediation framework, and potential biases such as collider stratification and residual confounding are explicitly discussed.

The authors have appropriately contextualized their unexpected finding that ART-conceived neonates had a lower risk of NICU admission, noting that this contrasts with much of the existing literature and requires cautious interpretation. They offer plausible explanations, such as enhanced prenatal monitoring and embryo selection, while acknowledging the speculative nature of these mechanisms and the limitations of retrospective observational data.

The manuscript is generally well written and intelligible. Although a few minor grammatical and stylistic issues remain, they do not impede understanding. A final round of light language polishing would further enhance clarity.

The data availability and ethics statements are appropriate, and the study appears to comply with all ethical and reporting standards for human subjects research.

Overall, this manuscript represents a valuable and timely contribution to the literature on ART and neonatal outcomes. I recommend minor revision to address a few residual language issues, but no further substantive changes are required.

Reviewer #3: The authors have provided a detailed and thoughtful response to all reviewer comments, significantly enhancing the manuscript's clarity, methodological transparency, and scientific rigor. The unexpected finding regarding lower NICU admission risk in ART-conceived neonates is now better contextualized with a balanced discussion and acknowledgment of its speculative nature. Methodological concerns, such as the data collection timeline and limitations of propensity score matching, were addressed with appropriate clarifications. Improvements were made to the consistency of variable reporting, NICU admission criteria were explicitly defined, and the mediation analysis was explained in greater depth. Language and formatting issues were resolved through professional editing. Overall, the revisions have substantially improved the manuscript and it is now suitable for publication with no major issues remaining.

Reviewer #4: (No Response)

**Do you want your identity to be public for this peer review?** For information about this choice, including consent withdrawal, please see our Privacy Policy

Reviewer #1: **Yes: ** yongchuan1972@gmail.com

Reviewer #3: No

Reviewer #4: No

---

## [Editor Report · Acceptance letter]

PONE-D-25-15056R1

PLOS ONE

Dear Dr. Wang,

I'm pleased to inform you that your manuscript has been deemed suitable for publication in PLOS ONE. Congratulations! Your manuscript is now being handed over to our production team.

Kind regards,

on behalf of

Professor Ahmed Mohamed Maged

Academic Editor

PLOS ONE